# Acute Mild Pancreatitis Following COVID-19 mRNA Vaccine in an Adolescent

**DOI:** 10.3390/children9010029

**Published:** 2021-12-31

**Authors:** Ahmad Kantar, Manuela Seminara, Marta Odoni, Ilaria Dalla Verde

**Affiliations:** Paediatric Unit, Istituti Ospedalieri Bergamaschi, Gruppo Ospedaliero San Donato, 24036 Bergamo, Italy; manuela.seminara17@gmail.com (M.S.); odomarta@yahoo.it (M.O.); ilaria.dallaverde@gmail.com (I.D.V.)

**Keywords:** COVID-19, vaccine, pancreatitis

## Abstract

A 17-year-old male was referred to the emergency room with sharp abdominal pain, pallor, sweating, and vomiting 12 h after the administration of his first Pfizer-BioNTech vaccine for coronavirus disease 2019 (COVID-19). He had abdominal pain, an increase in serum lipase value of > 3 times the upper limits of normal, and magnetic resonance imaging (MRI) findings consistent with acute mild pancreatitis (AP). He was started on treatment with fluid therapy and non-steroidal anti-inflammatory drugs for pain management, after which he recovered rapidly and was discharged on the fourth day after hospitalization. The available data are difficult to interpret as AP is a relatively frequent disease, but its occurrence after vaccination seems extremely rare. Although it is a rare event, AP should be considered after COVID-19 vaccination, especially in those exhibiting abdominal tenderness and vomiting, which should be promptly treated and adequately investigated.

## 1. Introduction

Vaccines are considered one of the most important public health achievements of the last century. Depending on the disease, the biology of the infection, and the target population, a vaccine may require the induction of different adaptive immune mechanisms to be effective. The concept of nucleic acid-encoded vaccines was conceived three decades ago when Wolff et al. demonstrated that direct injection or “naked delivery” of messenger RNA (mRNA) or plasmid DNA (pDNA) into the skeletal muscle of a mouse resulted in expression of the encoded protein [1].

This seminal research appeared promising, although challenges in regard to working with mRNA has raised many concerns for its therapeutic applications. This is mainly because mRNA produces only low levels of proteins and degrades quickly. In addition, mRNA may produce an immune reaction that is independent of the response to the protein that it encodes. Years later, Karikó et al. addressed major limitations and demonstrated that synthetic nucleosides could increase protein production from mRNA and drastically suppress the immune system’s reaction to the mRNA molecules themselves [2]. Protective immune responses by mRNA vaccines against infectious pathogens was demonstrated a decade ago [3].

The outbreak of severe acute respiratory syndrome coronavirus 2 (SARS-CoV-2) that began in Wuhan has rapidly evolved into a global pandemic [4]. This has prompted researchers in the field of vaccines to join with pharmaceutical companies to focus all resources on promptly developing anti-SARS-CoV-2 vaccines. Months later, the RNA vaccine platform has enabled rapid vaccine development in response to this pandemic.

Pfizer-BioNTech developed the vaccine BNT162b1, a lipid nanoparticle-formulated nucleoside-modified mRNA that encodes the receptor binding domain (RBD) of the SARS-CoV-2 spike protein. The vaccine RNA is formulated in lipid nanoparticles for very efficient delivery into cells after intramuscular injection [5]. The vaccine has been employed worldwide in the fight against SARS-CoV-2 infection in subjects aged 12 years and over.

In expansive vaccination campaigns with a newly released vaccine, it is normal to identify potential adverse events following immunization. These findings do not necessarily mean that the events are linked to vaccination itself, but they must be investigated to guarantee that any safety concerns are adequately addressed. We report a case of acute mild pancreatitis (AP) in a 17-year-old adolescent following the administration of his first dose of Pfizer-BioNTech vaccine for coronavirus disease 2019 (COVID-19) and review our current knowledge of similar cases. Written informed consent was obtained from the caregivers of the patient and the patient himself as indicated by the ethics committee.

## 2. Case Presentation

A 17-year-old male was referred to the emergency room with sharp abdominal pain, pallor, sweating, and vomiting 12 h after the administration of his first Pfizer-BioNTech COVID-19 vaccine. He had no diarrhoea or fever. He suffered from allergic rhinitis but was undergoing treatment on demand with topical steroids (budesonide nasal spray). He had no history of infections, fever, or contact with a sick person prior to vaccination. No recent dietary changes or alcohol consumption was reported. He reported a family history of diabetes (his father has type I diabetes).

On physical examination, his condition was discrete with pallor, and he had epigastric tenderness with pain irradiating to the spine, which he described as 10/10 on a pain scale. Vital signs were within the normal range with a heart rate of 80 beats per minute, arterial blood pressure of 115/75 mmHg, respiratory rate of 20 breaths per minute, and oxygen saturation of 97%. Laboratory analysis revealed positive serum biomarkers for pancreatitis: (1) serum lipase at 1535 U/L (normal range: 70–390 U/L) and (2) serum amylase at 161 U/L (normal range: 25–115 U/L). Laboratory tests included serum cholesterol, triglyceride, electrolytes, calcium, alkaline phosphate, alanine aminotransferase, aspartate aminotransferase, gammaglutamyl transferase, bilirubin, blood urea nitrogen, creatinine, complete blood count, blood eosinophils, tryptase, D-dimer, troponin I, N-terminal pro B-type natriuretic peptide, procalcitonin, and C-reactive protein (Table 1).

Reverse transcriptase polymerase chain reaction (RT-PCR) tests for SARS-CoV-2 from nasopharyngeal swabs were negative on two occasions. Blood and urine tests for alcohol and drug abuse and chest X-rays were negative on hospital admission. A quantitative IgG antibody test for SARS-CoV-2 was negative upon hospital admission. Echocardiogram performed on the patient on second day of admission was normal. Abdominal magnetic resonance imaging (MRI) demonstrated pancreatic enlargement and increased intensity due to oedema with inflammation surrounding the pancreas (Figure 1). Liver ultrasound imaging excluded the presence of gallbladder sludge or an abnormal biliary tree. Urine analysis and renal ultrasound imaging demonstrated the absence of renal involvement. His weight was 61 Kg, height 168 cm and BMI 24 Kg/m^2^.

AP was diagnosed based on the recommendations of the North American Society for Pediatric Gastroenterology, Hepatology, and Nutrition Pancreas committee [6], since the patient presented abdominal pain compatible with AP, an increase in serum lipase value of >3 times the upper limits, and abnormal MRI imaging findings. The patient was placed on IV fluid therapy consisting of lactated Ringer’s solution (Fresenius Kabi, Verona, Italy) to correct hypovolemia and maintain adequate fluid status and urine output. He was also given non-steroidal anti-inflammatory drugs: paracetamol (Fresenius Kabi, Verona, Italy) 1 g three times daily for two days, and indomethacin (Alfasigma, Pomezia, Italy) 25 mg 1 cps twice daily on day one for pain management. Oral nutrition was started 48 h after presentation.

The overall outcomes were favourable, and serum biomarkers for pancreatitis were normal on the fourth day. The patient was discharged from the hospital. Four weeks later, observations revealed an absence of islet cell antibodies and antibodies for glutamic acid decarboxylase (GAD-65) and t-transglutaminase, whereas a quantitative IgG antibody test for SARS-CoV-2 was positive. The abdominal MRI was normal (Figure 2). A second mRNA COVID-19 vaccine dose was not administered to the adolescent.

## 3. Discussion

The Pfizer-BioNTech COVID-19 vaccine represents a significant contribution to the ongoing vaccination campaign. Although the vaccine has been proven to be effective and safe, vaccine-induced side effects have been observed. Among the most frequently reported gastrointestinal side effects are nausea, diarrhoea, decreased appetite, abdominal pain, vomiting, heartburn, and constipation [7]. According to Pfizer’s data, one case of pancreatitis and one of obstructive pancreatitis adverse reaction were observed during the phase 2/3 clinical trial of the COVID-19 mRNA vaccine [8]. The trial included about 38,000 participants, indicating that such a link between vaccination and pancreatitis is a very rare adverse reaction.

Using information obtained from UK databases, spontaneous reports received between 9 December 2020 and 21 July 2021 were collected for the mRNA Pfizer/BioNTech vaccine. The collection contains 275,820 adverse reaction reports, which include one case of obstructive pancreatitis, nine cases of pancreatitis, nine cases of AP, and one case of necrotising pancreatitis [9]. The National Agency for the Safety of Medicines and Health Products (L’Agence Nationale de Sécurité du Médicament et des Produits de Santé [ANSM]) reported severe side effects of the Pfizer-BioNTech vaccine for up to 1 July 2021. There were 57 cases of pancreatitis out of a total of 42,523,573 doses (first injection *N* = 26,142,447; second injection *N* = 168,126) [10]. Data have also been released from VigiBase, the World Health Organization’s (WHO’s) global database of individual case safety reports from cases involving the pancreas [11]. These cases include pancreatitis (313), AP (298), chronic pancreatitis (9), necrotizing pancreatitis (17), haemorrhagic pancreatitis (7), pancreatic cyst (8), pancreatic failure (5), pancreatic enlargement (4), pancreatic infarction (3), pancreatic pseudocyst (3), pancreatic atrophy (2), pancreatic duct dilatation (2), pancreatic haemorrhaging (2), pancreatic steatosis (2), pancreatitis relapse (2), pancreatic duct stenosis (1), pancreatic enzyme abnormality (1), pancreatic atrophy (2), pancreatic haemorrhage (2), pancreatitis relapse (2), and pancreatic duct stenosis (1).

Single cases of pancreatitis after a Pfizer-BioNTech COVID-19 vaccine have been reported in the literature. Parkash et al. reported a case of AP in a 96-year-old female a few days after she received the vaccine [12]. Walter et al. reported a case of acute necrotizing pancreatitis 10 h after the administration of the second dose of Pfizer-BioNTech COVID-19 mRNA vaccine in a 43-year-old male [13]. Cieslewicz reported a case of AP injury 20 h after Pfizer-BioNTech COVID-19 mRNA vaccination in a 29-year-old female [14]. As in the latter two cases, our patient presented mild AP shortly after receiving the vaccination without signs or symptoms of an allergic reaction.

Viral vaccine-induced pancreatitis is an extremely rare adverse reaction that has been reported in single cases in the literature for mumps, measles, rubella [15,16,17], hepatitis A and B [18], and human papillomavirus [19]. Currently, the association between COVID-19 vaccination and AP is not based on evidence. Furthermore, emerging data are difficult to interpret as AP is a relatively frequent disease, and its occurrence after vaccination seems extremely rare. Although it is difficult to draw conclusions about the likelihood of the vaccine being an etiological factor for AP, immune and inflammatory responses may have precipitated this event. Nonetheless, our case presented mild pancreatitis with rapid recovery.

AP should be considered after COVID-19 vaccination, especially in those exhibiting abdominal tenderness and vomiting, which are symptoms that should be promptly investigated. The benefits of vaccination against COVID-19 significantly exceed the possible risks. An online registry for data collection allows for tracking the side effects of COVID-19 vaccines and increases our awareness of these side effects, as in the case of myocarditis in adolescents. Now that mRNA vaccines have revealed their potential, many more vaccine makers will likely foster an interest in mRNA technology.

## Figures and Tables

**Figure 1 children-09-00029-f001:**
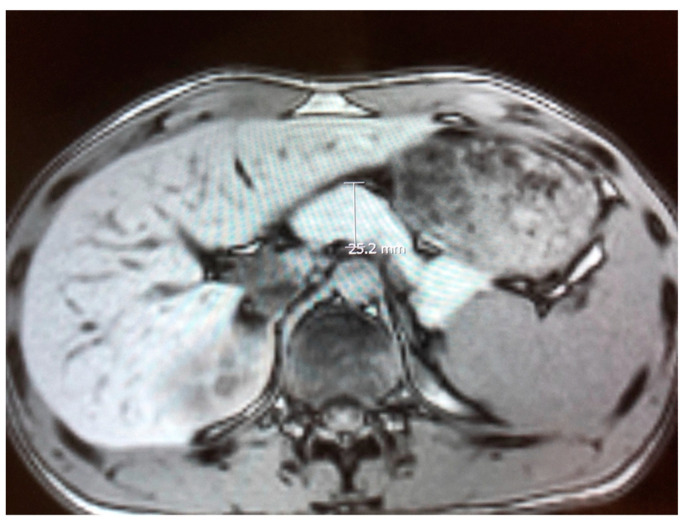
Abdominal magnetic resonance imaging showing enlargement of the pancreatic gland and an increase in its intensity.

**Figure 2 children-09-00029-f002:**
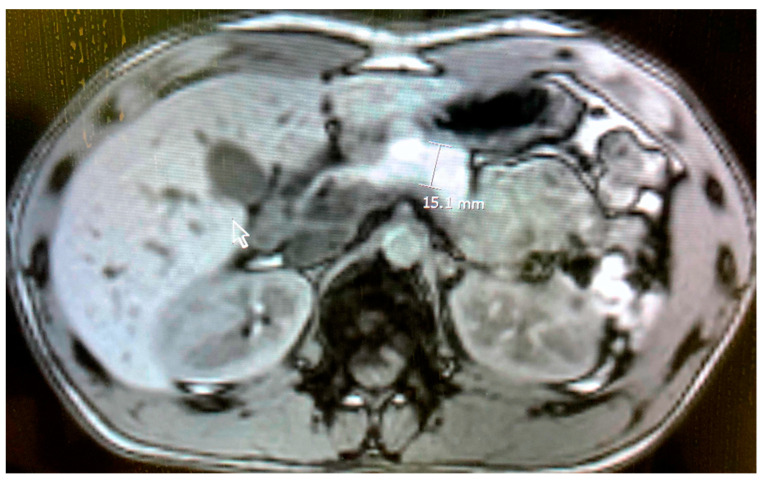
Normalization of abdominal magnetic resonance imaging of the pancreatic gland after treatment.

**Table 1 children-09-00029-t001:** Laboratory analyses on hospital admission.

Parameter	Blood Level (Normal Range)
Lipase	1535 (70–390) U/L
Amylase	161 (25–115) U/L
Triglyceride	68 (90–129) mg/dL
Cholesterol	83 (100–200) mg/dL
Sodium	140 (136–145) mEq/L
Potassium	4.5 (3.4–5.3) mEq/L
Chloride	107 (95–112) mEq/L
Calcium	9.3 (8.5–10.1) mg/dL
Alanine aminotransferase	34 (6–43) U/L
Aspartate aminotransferase	27 (3–37) U/L
Gammaglutamyl transferase	37 (3–65) U/L
Bilirubin	0.86 (0.2–1.0) mg/dL
Alkaline phosphatase	79 (70–935) U/L
Urea nitrogen	24 (16–45) mg/dL
Creatinine	0.88 (0.67–1.17) mg/dL
Glucose	90 (74–106) mg/dL
White blood cells	9.3 × 10^9^/L (3.4–9.6 × 10^9^/L)
Neutrophils	4.48 × 10^9^/L
Lymphocyte	4.1 × 10^9^/L
Monocytes	0.4 × 10^9^/L
Eosinophils	0.3 × 10^9^/L
Basophils	0.1 × 10^9^/L
Red blood cells	5.22 × 10^12^/L (4.35–5.65 × 10^12^/L)
Haemoglobin	15.5 (13.2–16.6) gr/dL
Platelets	188 × 10^9^/L (135–317 × 10^9^/L)
Tryptase	4 (<11) ug/L
D-dimer	<0.5 (<0.5) ug/mL
Troponin I	<0.015 (<0.015) ng/mL
N-terminal pro B-type natriuretic peptide	<5 (5–363) pg/mL
Procalcitonin	<0.05 (<0.05) ng/mL
C-reactive protein	<0.29 (<0.29) mg/dL

## Data Availability

Data sharing is not applicable to this article as no datasets were generated or analyzed during the current study. Data are available from the corresponding author upon request.

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
