# Peer review of "Acute Mild Pancreatitis Following COVID-19 mRNA Vaccine in an Adolescent"

_children, 2021, doi:10.3390/children9010029_

Round 1
Reviewer 1 Report
This is an interesting case of acute pancreatitis after COVID vaccine. It is mentioned that patient was undergoing treatment for allergic rhinitis, however no medication was specified. In fact, patient's medication list or prior history of pancreatitis was not mentioned. Please include them in the manuscript
Author Response
Dear reviewer,
thank you for your kind suggestions.
We have included the medications used in treatment of allergic rhinitis.
Reviewer 2 Report
Manuscript ID: children-1438665
Title: Acute mild pancreatitis following Covid-19 mRNA vaccine in an adolescent
Authors Ahmad Kantar et al.
The manuscript presents a case report of the patient with acute pancreatitis that developed after vaccination against COVID-19. The manuscript is interesting but contains some deficiencies and errors that need to be corrected. All responses to the reviewer’s comment should be presented in the next version of the manuscript.
List of deficiencies and errors:
- Section 2. Case presentation. The authors should provide more details about the patient’s condition on admission and discharge from the hospital. What was the heart rate, arterial blood pressure, respiratory rate, blood gas analysis and hemoglobin oxygen saturation in peripheral tissues?
- The authors stated that “no dietary changes or alcohol consumption was reported”. Have the authors tested the patient for the presence of alcohol or cannabinoids? Consuming alcohol before the age of 18, as well as taking drugs is illegal, therefore doctors should rely on tests, not on patient’s declarations. Alcohol and gallstones are the main factors responsible for the development of acute pancreatitis (PMID: 18853993). The authors did not report whether the patient was examined for the presence of gallstones disease? This should be clearly stated. There are also clinical (PMID: 32675788) and experimental (PMID: 18593574) evidence that cannabinoids can induce and/or worsen acute pancreatitis. This problem should be presented in the discussion.
- Section 2. Case presentation. What was the patient’s body weight, height, and body mass index? The authors stated that echocardiographic examination was performed. When was it curried out? What were the heart rate, stroke volume, end diastolic volume and cardiac output during this examination? What was the temperature of extremities? Pallor and sweating suggest a reduction in arterial blood pressure.
- Section 2. Case presentation, paragraph 3, the first sentence of this paragraph. The authors report a long list of tests performed, but do not present the results. The results of all tests should be presented in the form of Table 1.
- Case presentation, page 3, the penultimate paragraph of section 2. The authors wrote that “the patient was placed on IV fluid therapy consisting of lactated Ringer to correct hypovolemia and maintain adequate fluid status and urine output”. On the other hand, there is no data on the diuresis values during patient’s stay in the hospital. Were the actual and not estimated glomerular filtration rate and serum or urine neutrophil gelatinase-associated lipocalin (NGAL) tested? NGAL is well known marker of acute kidney injury (PMID: 25868473), as well as is postulated to be a prognostic biomarker in the early assessment of acute pancreatitis (PMID: 28013317). This problem should be presented in the discussion.
- Discussion. The reviewer does not know what the criteria were for linking COVID-19 vaccination with acute pancreatitis were, and the authors should check this and these criteria in the manuscript. Logically, such an association would have occurred if symptoms of acute pancreatitis had appeared within one week after vaccination. The global annual incidence of acute pancreatitis has increased over the past two decade, reaching a maximum of 85.4 cases per 100,000 people (PMID: 27161172). Therefore, the maximum weekly incidence of acute pancreatitis in approximately 1.64 per 100,000 inhabitants (data based on data in the article, PMID: 27161172). If the weekly incidence of acute pancreatitis after vaccination with COVID-19 is higher, it would indicate that vaccination may lead to the development of this disease.
- Discussion. The authors report that the patient suffered from allergic rhinitis. This indicates that the patient’s immune system overacts and abnormally reacts to exposure to antigens. The information that the patient’s father suffers from type I diabetes also indicates possible genetic condition that may lead to an abnormal and excessive immune reaction. Therefore, it is likely that primary possible cause of acute pancreatitis presented in this case report was an anaphylactic reaction to any compound of the vaccine or to any other antigen ingested. The pale skin and sweating observed on admission to the hospital also support this concept. In anaphylactic reaction, the blood pressure is lowered and, to maintain adequate blood flow through the heart and brain, blood flow to other organs is reduced, especially the digestive system and kidney. On the other hand, there are evidence that in pancreatitis, regardless of primary cause of this disease, there is a decrease in pancreatic blood flow (PMID: 14726612; PMID: 1996652). In addition, disturbance in pancreatic blood flow may be the primary cause of acute pancreatitis (PMID: 11453102; PMID: 686887; PMID: 10669996). This concept should be presented in the discussion. Moreover, the authors should consult the case with allergologists and consider advising the patient to refrain further vaccination against the coronavirus.
- There are some typing errors that need to be corrected, for example line 38, 39, 102, 120.
Author Response
Dear reviewer,
Thank you for your precious observations.
- Heart rate 78, Arterial blood pressure 120/75, Respiratory rate 30, saturation 97%. Similar data at discharge.
- The majority of adults are tested in our emergency room for alcohol and drug abuse among which those with gastrointestinal symptoms. The reported patient was negative. Abdominal ultrasound excluded the presence of gallbladder sludge and demonstrated normal biliary tree. This was added to text.
- Weight 61 height 168 BMI 24 . Echocardiographic examination was carried on day 2 of hospitalization. Normal heart volume and output were observed.
- We reported only the abnormal results, describing normal values does not offer the reader important data.
- The patient was treated as indicated by guidelines on acute pancreatitis. Renal function parameters and urine were in normal range. Additionally, abdominal ultrasound excluded renal involvement. Hence, NGAL and glomerular filtration rate were not carried in our case of mild pancreatitis.
- Our article and similar articles previously published do not demonstrate a link between vaccination and the onset of pancreatitis. This is beyond the scope of a case report.
- The patient did not present clinical symptoms or signs of anaphylactic reaction. Moreover, cases of allergic reactions to mRNA COVID-19 vaccines presented with different clinical symptoms (JAMA Network Open. 2021;4(8):e2122255., Am J Transplant. 2021;21:1332–1337. JAMA Network Open. 2021;4(9):e2125524. Blood parameters of allergy (peripheral eosinophils, tryptase were normal in our patient). This was added to text. The timing of onset of pancreatitis in our case and in other reported cases remains a brainteaser.
- The article was revised by an English editing agency.
Reviewer 3 Report
I would provide the following comments and suggestions:
1) The authors only showed result for patient MRI report indicating the abnormal pancreas. However, it would be more helpful if the author would show results of MRI from a negative control patient ( with vaccine but no symptom) or of a normal healthy patient.
2) The author reported that the MRI was normal four weeks later. It would be better if the authors could include that MRI data for clarification. Did the author repeated the MRI and other tests for pancreatitis 1/2 weeks after COVID detection.
3) Did the patient received 2nd dose of COVID vaccine? If yes, is the abnormality in pancreas observed for 2nd dose as well?
4) The author didn't reported whether a neutralizing antibody (Ab) test for serum samples of patient has been conducted or not. A neutralizing Ab test results with abnormal pancreas as well as with normal pancreas ( after recovery) should shed some light on the role of the vaccine and its effect for abnormal pancreatitis.
5) The authors also did not reported any results for the levels of key cytokines during the observed abnormal pancreatitis after vaccination.
Author Response
Dear reviewer,
thank you for kind and useful suggestions.
1-2) We have added to the article abdominal MRI after treatment. Blood parameters were normalized. We have investigated the presence of pancreas antibodies, that were negative.
3) Second mRNA COVID-19 vaccine dose was not administered to the adolescent. This was added to the text.
4) Unfortunately, no investigation of cytosine level was carried
Reviewer 4 Report
The presented case is interesting and would gain attention from physicians and scientists. The title is informative and relevant. The references are relevant and recent. The cited sources are referenced correctly. Appropriate and key studies are included.
The case is well-described, the used methods for diagnosing and therapy are valid and reliable. The patient data is presented in an appropriate way.
Data is discussed from different angles and placed into context without being overinterpreted.
The conclusions are supported by references and own results.
This paper added to what is already in the topic. The article is consistent within itself.
Specific comments on weaknesses of the article and what could be improved:
Major points -
1. The conclusion regarding that the causative relationship between vaccination and AP development is not proved. Therefore, after vaccination, we should be aware of this condition, but not expect it. We have to be extremely careful when discussing this sensitive topic and to choose the proper words.
Minor points
1. It is convenient to mention that the overall prognosis of AP is good, and to follow up the boy outcomes.
Author Response
Dear reviewer,
thank you for your kind observation.
The aim of the study is to note this case report and to review similar cases reported in literature. No collected data demonstrated a causative relationship between vaccination and AP development.
We added to the text that our case presented mild pancreatitis with rapid recovery.
Round 2
Reviewer 2 Report
Manuscript ID: children-1438665
Title: Acute mild pancreatitis following Covid-19 mRNA vaccine in an adolescent
Authors Ahmad Kantar et al.
The new version of the manuscript is almost ready for publication. However, there are still some minor errors and deficiencies that should be corrected.
List of deficiencies and errors:
- Line 78-79. The author should add units of presented results.
- Line 82-86. The authors present a list of laboratory tests performed, but do not provide their results. The best way to present the results would be a separate table. In response to the reviewer, the authors stated that there is no such need because the results were within the normal range. However, even this information is missing from the manuscript. Moreover, at the present stage of case knowledge, these results may turn out to be important and should therefore be presented.
- For the reviewer’s third comment on the case description, the authors responded that “Weight 61 height 168 BMI 24. Echocardiographic examination was carried on day 2 of hospitalization”. These data with appropriate units should be presented in the manuscript.
- In the case of drugs and fluids used, the authors should provide the name of these agents (commercial and generic0, their source (manufacturer, city, and country name) and the doses used.
Author Response
Thank you for your precious comments.
The following modifications were added to the text
- As requested the units were added (lie 78-79)
- A table of laboratory tests was added
- Body weight, height and BMI were added
- Name of drugs used were added.
Reviewer 3 Report
According to me, the manuscript has been sufficiently improved.
Author Response
Thank you for your kind revision.
Reviewer 4 Report
The authors addressed all the issues raised by the reviewers. The paper has been improved significantly. No further comments.
Author Response
Thank you for your kind revision.